# Ginsenoside CK Inhibits the Early Stage of Adipogenesis via the AMPK, MAPK, and AKT Signaling Pathways

**DOI:** 10.3390/antiox11101890

**Published:** 2022-09-23

**Authors:** Jung-Mi Oh, Sungkun Chun

**Affiliations:** 1Department of Physiology, Jeonbuk National University Medical School, Jeonju 54907, Korea; 2Institute of Medical Sciences, Jeonbuk National University Medical School, Jeonju 54907, Korea; 3Research Institute for Endocrine Sciences, Jeonbuk National University Medical School, Jeonju 54907, Korea

**Keywords:** compound-K, adipogenesis, obesity, ROS, AMPK, differentiation, ERK/P38 signaling

## Abstract

Obesity is considered a health hazard in part due to the associated multiple diseases. As rates of obesity continue to increase, a new strategy for its prevention and treatment is required. Compound-K, an active ingredient in ginseng, possesses antioxidant, anti-inflammatory, and anti-cancer properties. Although ginseng has used as various therapeutics, its potential ability to alleviate metabolic diseases by regulating adipocyte differentiation is still unknown. In this study, we found that CK treatment significantly inhibited lipid droplet and adipogenesis by downregulating the mRNA expression of C/ebpα, Ppar-γ, Fabp4, Srebp1, and adiponectin as well as protein levels of C/EBPα, PPAR-γ, and FABP4. CK also decreased the production of reactive oxygen species (ROS), while it increased endogeneous antioxidant enzymes such as catalase, glutathione peroxidase (GPx), glutathione reductase (GR), superoxide dismutase (SOD) 3 and SOD2. We observed that CK treatment suppressed the expression of cyclin-dependent kinase 1 (CDK1) and cyclin B1 during the mitotic clonal expansion (MCE) of adipocyte differentiation, and it arrested adipocytes at the G2/M stage due to the increased expression of p21 and p27. CK decreased the phosphorylation of extracellular signal-regulated kinase (ERK) and p38 and protein kinase B (AKT) in early-stage adipogenesis. In addition, the inhibition of adipogenesis by CK significantly increased the phosphorylation of AMP-activated protein kinase (AMPK) and acetyl-CoA carboxylase (ACC). Interestingly, AMPK pharmacological inhibition with Dorsomorphin limited the effect of CK on suppressing PPAR-γ expression in differentiated 3T3-L1 cells. Our results suggest that CK exerts anti-adipogenic effects in 3T3-L1 cells through the activation of AMPK and inhibition of ERK/p38 and AKT signaling pathways.

## 1. Introduction

Obesity, a complex disease that causes an excessive accumulation of fat, is increasing in frequency worldwide and is widely regarded as one of the most serious health problems [1]. Obesity has been reported as a risk factor for the development of other metabolic syndromes, such as dyslipidemia, hypertension, diabetes, and cardiovascular disease [2,3,4]. 

Obesity is characterized by an increase in the number (hyperplasia) and size (hypertrophy) of white adipocytes (WATs), which are induced as a result of adipogenesis, i.e., the process of adipocyte differentiation [4,5]. The increased storage of triacylglycerol by the endogenous lipogenic pathway expands the size of adipocytes, and the number of adipocytes can be increased by inducing the proliferation as well as differentiation of pre-adipocytes [6]. Thus, inhibiting the differentiation of pre-adipocytes to adipocytes, reducing lipogenesis, increasing lipolysis, and inducing apoptosis in mature adipocytes are important processes of preventing and managing obesity.

Many studies have linked oxidative stress in adipocytes to obesity and type 2 diabetes [7,8]. Oxidative stress is determined by the balance of antioxidants and reactive oxygen species (ROS) [9]. The production of ROS in white adipose tissue affects the endocrine and metabolic functions of adipocytes. In most normal cells, optimal redox conditions are achieved by low levels of ROS and high antioxidant levels [8]. In contrast, obesity exhibits higher levels of oxidative stress in WAT, including increased ROS levels and decreased activity of endogenous antioxidant enzymes such as catalase, SOD, GR and GPx [10]. Increased oxidative stress is also associated with abdominal obesity and insulin resistance [10,11].

3T3-L1 is a well-known murine pre-adipocyte cell line that is widely used for in vitro experiments in obesity studies [12,13]. The adipogenic process in 3T3-L1 cells is divided into four stages, including: (1) growth arrest, (2) MCE, (3) early differentiation, and (4) late differentiation [14]. In the early stage of differentiation, quiescent cells are re-entering the cell cycle and undergoing MCE. As MCE is an essential step in the early stage of pre-adipocyte differentiation, a malfunction at this stage can lead to serious problems in the adipogenesis process [15]. The CDK complex, which includes the cell cycle regulators cyclin D, E, A, and B, is essential for retinoblastoma (Rb) phosphorylation and cell cycle re-entry. CDK inhibitors such as p21CIP and p27KIP1 are associated with cell cycle-arrested pre-adipocytes, whereas the degradation of them results in G1/S phase progression.

During adipogenesis, precursor adipocytes transform into mature adipocytes through a multi-step process involving the expression of various transcription factors and signaling pathways [16]. The expression of CAAT/enhancer binding protein ß (C/EBPß) increases during the early stages of adipogenesis, including the MCE phase. Elevated C/EBPβ promotes the expression of C/EBPα as well as peroxisome proliferator-activated receptor γ (PPAR-γ) transcriptional initiation [4]. PPAR-γ and C/EBPα are known to upregulate triglyceride synthesis and the expression of several genes, including ACC, adipocyte fatty acid binding protein 2 (aP2, also known as FABP4), and fatty acid synthase (FAS), as well as sterol regulatory element binding protein 1c (SREBP1c) in the late stages of adipogenesis [17,18]. As C/EBPα and PPAR-γ are the key regulators of adipogenesis, they determine the expression of multiple genes that are responsible for fat accumulation and insulin sensitivity. Adipocyte differentiation is facilitated via crosstalk between adipogenic transcription factors (C/EBPα, C/EBPß, C/EBPδ and PPAR-γ) and cell cycle regulators (CDKs). Therefore, interfering with cell cycle progression or suppressing the expression of adipogenic transcription factors during MCE may be one method of inhibiting adipogenesis.

Adenosine monophosphate-activated protein kinase, a type of serine/threonine protein kinase, is activated by cellular stresses which deplete adenosine triphosphate (ATP) [19]. AMPK is a metabolic master protein that controls the synthesis and degradation of fatty acids (FA), and it plays a key role in maintaining the homeostasis of cellular energy [20,21]. AMPK activation is achieved by phosphorylation at the threonine residue (T172) of the α-subunit [22], which inhibits the de novo synthesis of fatty acids (FA), cholesterol, and triglycerides (TG) as well as promotes catabolic pathways such as fatty acid oxidation. Activated AMPK attenuates lipid accumulation and sterol synthesis by inhibiting the phosphorylation of ACC1, SREBP1c, and FAS [22,23]. Pre-adipocyte differentiation can be suppressed via AMPK activation and by the repression of key transcription factors of adipogenesis, such as C/EBPα, PPAR-γ, and SREBP-1c [24,25]. Therefore, the activation of AMPK is an important factor in the treatment of obesity. 

Panax ginseng, which has been used for medicinal purposes for thousands of years in China, Korea, and Japan, contains the pharmacologically active component ginsenoside. Ginsenosides are generally divided into protopanaxadiol (PPD)-type and protopanaxatriol (PPT)-type according to the number of hydroxyl groups that can be removed from the sugar moiety through dehydration [26]. A rare ginsenoside Compound K (CK) is one of the intestinal metabolites of 20(S)-PPD derivatives and exhibits strong anti-inflammatory, anti-cancer, anti-diabetic and hepato-protective effects [27,28]. 

Ginsenoside CK increases the level of p-AMPK and inhibits TAG synthesis in 3T3-L1 cells [29]. Ginsenoside CK has been shown to inhibit adipogenesis in 3T3-L1 cells by suppressing C/EBPα and PPAR-γ [30]. Recently, Wang et al. (2021) reported that ginsenoside CK improves insulin resistance in obesity by inhibiting macrophage activity [28]. However, there are only a few studies of CK on adipogenesis, and the detailed molecular mechanism of CK on adipogenesis in 3T3-L1 cells is not yet fully understood. Therefore, in this study, we aimed to investigate CK’s effect on the stages of adipocyte maturation as well as the signaling pathways involved in this process.

## 2. Materials and Methods

### 2.1. Antibodies and Other Reagents

CK (purity > 97%) was prepared by modifying PPD-type ginsenosides that were extracted from Korean ginseng via acid-heat treatment. CK was prepared in 100% dimethyl sulfoxide (DMSO), stored in small aliquots at −80 °C, and diluted in the cell culture medium as needed. Dulbecco’s Modified Eagle Medium (DMEM), bovine calf serum (BCS), fetal bovine serum (FBS), 0.05% Trypsin–EDTA, and penicillin–streptomycin were purchased from Gibco BRL (Grand Island, NY, USA). Insulin, dexamethasone (DEX), 3-isobutyl-1-methylxanthine (IBMX), rosiglitazone, Oil Red O, Hoechst 33342, protease and phosphatase inhibitor cocktail, DMSO, N-acetyl-L-cysteine (NAC), and 2’, 7’-dichlorodihydrofluorescein diacetate (DCFH-DA) were obtained from Sigma Chemical Co. (St. Louis, Mo, USA). Antibodies for the following proteins were purchased from Cell Signaling Technology (Danvers, MA, USA): P38MAPK, phospho (p)-P38MAPK (Thr180/Tyr182), AKT, p-AKT, ERK1/2, p-ERK1/2, AMPKα, p-AMPKα, Acetyl-CoA Carboxylase (ACC), p-ACC, peroxisome proliferator-activated receptor gamma (PPAR-γ), CCAAT/enhancer-binding protein α (C/EBPα), fatty acid-binding protein 4 (FABP4), p21, p27, CDK1, cyclin B1, catalase, and superoxide dismutase (SOD)2. Glutathione peroxidase (GPx), glutathione reductase (GR), and SOD3 was purchased from Santacruz Biotechnology Inc. (Dallas, TX, USA), and the glyceraldehyde-3-phosphate dehydrogenase (GAPDH) antibody was purchased from Bioworld Technology (St. Louis Park, MN, USA).

### 2.2. Cell Culture and Adipocyte Differentiation 

Differentiated 3T3-L1 adipocytes are a widely used in vitro model of white adipocytes [31]. 3T3-L1 mouse pre-adipocytes were purchased from the American Type Culture Collection (ATCC: Manassas, VA, USA). 3T3-L1 pre-adipocytes were cultured in high-glucose Dulbecco’s Modified Eagle Medium (DMEM) supplemented with 10% (*v*/*v*) heat-inactivated fetal calf serum (FCS) and 1% (*v*/*v*) penicillin–streptomycin (P/S) in a humidified incubator with 5% CO_2_ atmosphere, at 37 °C, until confluent. After 2 days of confluence, pre-adipocytes were stimulated in differentiation medium (DMI) with DMEM that contained 10% FBS, 1% P/S, 0.5 mM IBMX, 1 μM dexamethasone, 1 μM rosiglitazone, and 10 μg/mL insulin for 2 days. After 2 days, the differentiation medium was replaced with DMEM that contained 10% FBS, 1% P/S and 1 μg/mL insulin, which was replenished every other day, for 8 days. Cells were fully differentiated into mature adipocytes by day 8. To evaluate the effect of CK on adipogenesis, pre-adipocytes and adipocytes were treated with CK and vehicle (DMSO) according to the respective conditions detailed in each figure’s legend. 

### 2.3. Cell Viability Assay

Cell viability was assessed using the Cell Count Kit-8 assay (CCK-8: Dojindo Molecular Technologies, Tokyo, Japan). 3T3-L1 pre-adipocytes (2 × 10^4^ cells/well) were seeded in a 96-well plate and treated with various concentrations of CK (0, 10, 20, 40, 60, and 80 µM) for 24 h at 37 °C under 5% (*v*/*v*) CO_2_. The medium was then removed, and 10 μL of the CCK-8 solution was added to each well of the plate for 2 h. The absorbance at 450 nm was measured using a Synergy™ microplate reader (BioTek Instruments Inc., Winooski, VT, USA). Cell viability was expressed as a percentage of the control, which was set to 100. All experiments were performed in triplicate and repeated at least three times.

### 2.4. Cytotoxicity Assay

The effect of CK on the viability of 3T3-L1 adipocyte cells was analyzed using the Trypan blue assay. After the differentiation of 3T3-L1 pre-adipocytes and CK treatment processes were completed, as described above, the cells were collected, stained with Trypan blue, and the number of cells was measured under a microscope (Leica DM IL LED, Leica Microsystems GmbH, Wetzlar, Germany) using a Luna cell counting slide (Logos Biosystems, Anyang, Korea).

### 2.5. Oil Red O Staining and Quantification 

Oil Red O staining was performed to reveal the intracellular lipid content of adipocytes. Adipocytes cells that had been differentiated for 8 days were rinsed 3 times with PBS and then fixed with 10% formaldehyde for 20 min. After the cells were reacted with 60% isopropanol for 5 min, 0.5% Oil Red O (0.5 g in 100 mL of isopropanol) was diluted with H_2_O at a ratio of 60:40 (*v*/*v*) Oil Red O/H_2_O, and the cells were kept at room temperature for 2 h. Finally, the Oil Red O solution was removed, and the cells were immediately washed four times with distilled water. The Oil Red O stained cultures were photographed using an inverted laboratory microscope (Leica DM IL LED, Leica Microsystems GmbH, Wetzlar, Germany) at a magnification of 200× (Leica, Germany). To quantify lipid accumulation, culture dye stained with Oil Red O was eluted with 100% isopropanol for 30 min, and absorbance at 510 nm was measured using a microplate reader (Synergy™, BioTek Instruments Inc., Winooski, VT, USA).

### 2.6. RNA Isolation and Quantitative Reverse Transcription (qRT)-PCR

Total RNA was extracted using TRIZOL reagent (Invitrogen, Carlsbad, CA, USA) according to the manufacturer’s protocol, and 5 µg of total RNA was used for complementary DNA (cDNA) synthesis using the GoScript™ Reverse Transcription System (Promega, Madison, WI, USA). For quality control, RNA purity and integrity were evaluated using a DS-11 FX UV-Vis spectrophotometer (DeNovix Inc., Wilmington, DE, USA). RNA was then stored at −80 °C prior to real-time quantitative PCR (RT-qPCR) analysis. For quantitative real-time PCR (qPCR), cDNA was mixed with Luna^®^ Universal qPCR Master Mix (New England Biolabs, Ipswich, MA, USA) and specific primers. The qRT-PCR reaction was processed using a magnetic induction cycler (MIC) PCR machine (Bio Molecular Systems, Australia). The relative expression level of a target gene was quantified by normalization with the internal control GAPDH gene, and expression difference was calculated according to the 2^−ΔΔCT^ method. The qRT-PCR primer sequences used in this study are listed in Table 1. 

### 2.7. Western Blot Analysis

Cells were lysed with radioimmunoprecipitation assay (RIPA) buffer (Biosesang, Seongnam-si, South Korea) supplemented with Halt™ Protease and Phosphatase Inhibitor Cocktail (Thermo Scientific, Waltham, MA, USA) on ice. Lysates were sonicated and cleared by centrifugation at 12,000× *g* at 4 °C for 20 min. Total lysates (30 µg) were separated using sodium dodecyl sulfate-polyacrylamide gel electrophoresis (SDS-PAGE) and then transferred onto a polyvinylidene fluoride (PVDF) membrane (Millipore Corp., Burlington, MA, USA). The membrane was blocked with 5% milk or 5% Bovine Serum Albumin (BSA) for 1 h at room temperature (RT), incubated overnight with primary antibodies, and then treated with horseradish peroxidase (HRP)-conjugated secondary antibodies for 1 h at RT. The blot bands were visualized using Clarity Western ECL Substrate (Bio-Rad Laboratories, Hercules, CA, USA), and proteins were detected with an Amersham Imager 600 (GE Healthcare, Piscataway, NJ, USA). The band intensities were quantified using ImageJ software, version 1.52a (NIH, Bethesda, MD, USA).

### 2.8. Immunofluorescence Cell Staining

Pre-adipocytes 3T3-L1 cells were seeded on slides and cultured for 48 h. Confluent cells were treated with DMI and then cultured for 8 days, both with and without CK. Cells were washed twice with PBS, fixed with 10% formaldehyde, and then blocked with 0.2% Triton X-100 and 5% normal goat serum in PBS for 1 h. Thereafter, primary antibodies (C/EBPα and PPAR-γ) were incubated at 4 °C overnight, and Alexa 488-conjugated anti-mouse IgG or Alexa 594-conjugated anti-rabbit IgG was reacted at room temperature for 1 hour. Nuclei were counterstained using Hoechst 33342 (BD Biosciences, San Jose, CA, USA), and the stained cells were observed under a CELENA S fluorescence microscope (Logos Biosystems, Anyang, Korea). Images were analyzed using Image J software, version 1.52a (NIH, Bethesda, MD, USA).

### 2.9. Determination of Intracellular ROS Generation

ROS production was assessed using the dye DCFH-DA (Sigma, MO, USA) as previously described [32]. Briefly, 3T3-L1 pre-adipocytes were seeded on coverslips in 24-well plates and in clear-bottomed black 96-well plates, differentiated for 8 days, and then treated with 20 μM DCFH-DA. After incubation for 30 min in 5% CO_2_ incubator at 37 °C, cells were rinsed with cold PBS, and the stained cells were imaged under a fluorescence microscope (CELENA S, Logos Biosystems, Anyang, Korea). Fluorescence intensity was measured at an excitation/emission wavelength of 435 nm/535 nm using a microplate reader Synergy^TM^ (BioTek Instruments, Inc., Winooski, VT, USA), and the results were expressed as a fold change of ND.

### 2.10. BODIPY Staining of Neutral Lipid Droplets (LDs)

BODIPY 493/503 dye (Thermo Fisher Scientific, Waltham, MA, USA) was used for staining the neutral lipid droplets (LDs), and the staining method was performed according to the manufacturer and other previous reports [33]. Fully differentiated 3T3-L1 adipocytes were incubated for 15 min at 37 °C with PBS containing 2 μM BODIPY probe. Stained cells were harvested, washed twice with PBS, and LDs intensity was quantified using CytoFLEX flow cytometry (Beckman Coulter, Inc., Kraemer Blvd. Brea, CA, USA). LD morphology was observed using a CELENA S fluorescence microscope (Logos Biosystems, Anyang, Korea) after fixing for 30 min at room temperature with 4% paraformaldehyde (PFA).

### 2.11. Statistical Analysis 

Data were presented as the means ± standard deviation (SD) from at least three independent experiments. Statistical analysis was performed using GraphPad Prism 8.0 (GraphPad Software Inc., San Diego, CA, USA) and SPSS software (version 12.0, SPSS Inc., Chicago, IL, USA). Significant differences among multiple groups were analyzed using one-way factorial analysis of variance (ANOVA), which was followed by Tukey’s test. Statistical significance was established at *p* < 0.05 (*/*^#^*), *p* < 0.01 (**/*^##^*) and *p* < 0.001 (***/*^###^*). 

## 3. Results

### 3.1. CK Inhibits Lipid Accumulation during the Early Phase of 3T3-L1 Adipogenesis

To determine whether CK treatment affects the cell viability of 3T3-L1 pre-adipocytes, we performed a CCK-8 assay. We found that CK treatment, regardless of concentration (10–80 μM), had no effect on the viability of pre-adipocytes (Figure 1B). In our subsequent experiments, CK concentration ranged between 0 and 40 μM.

3T3-L1 cell lines were treated with CK (30 μM) during adipogenesis to investigate its effect on adipocyte differentiation, and intracellular lipid accumulation was investigated (Figure 1C). The differentiation of confluent (day 0) 3T3-L1 pre-adipocytes was triggered by adding differentiation-inducing cocktails (DMI: dexamethasone, IBMX, rosiglitazone, and insulin) in a medium containing DMEM and FCS. We observed three stages of 3T3-L1 pre-adipocyte differentiation that were induced using DMI: early (days 0–2), intermediate (days 2–4), and late (after day 4). After 8 days of differentiation, the cells were stained with Oil Red O (Figure 1D). We found that the size and number of stained lipid droplets were enlarged and increased in the DMI-treated differentiated group (control) than in the undifferentiated group (undiff.). Enlargement of size and increase in number of lipid droplets were inhibited by treating cells with CK (30 μM) during the early (A and B) and intermediate (C) stages of differentiation. However, during the later stages of differentiation, CK treatment slightly decreased the size of lipids and increased the number of lipid droplets (Figure 1E). To more accurately quantify the degree of suppression for lipid accumulation caused by CK, Oil Red O-stained cells were eluted with isopropanol, and absorbance was measured at 510 nm with a spectrophotometer (Figure 1E). Oil Red O staining revealed that triglyceride (TG) content was significantly higher in the control groups than in the undifferentiated groups. The increased lipid content in the control group was significantly decreased in the CK-treated group from the beginning of differentiation to the entire days (A), 0–2 days (B), and 2–4 days (C). However, groups treated with CK during the late stages of differentiation (D, E and F) did not exhibit significantly reduced lipid accumulation (Figure 1D,E). These results suggest that CK inhibits lipid accumulation during the early stages of adipocyte differentiation.

### 3.2. CK Inhibits the Expression of Adipogenic Marker Genes in Mature 3T3-L1 Adipocytes

We performed qRT-PCR, Western blot (WB), and immunocytochemistry (ICC) analyses to investigate the inhibitory effect of CK on adipogenic gene expression in mature 3T3-L1 cells. The differentiation of pre-adipocytes into mature adipocytes is regulated by transcription factors and genes including PPAR-γ, C/EBPβ, C/EBPα, and FABP4 [16]. While we found that mRNA expression of Ppar-γ, C/ebpα, Fabp4, and Srebp1c decreased in a dose-dependent manner in the 20, 30, and 40 μM CK-treated groups, 10 μM CK treatment did not have this effect compared to the differentiated group. Interestingly, adiponectin mRNA expression was also downregulated in the CK treatment group (Figure 2A). As shown in Figure 2B–E, CK significantly decreased protein expression levels of adipogenic genes, including FABP4, C/EBPα, and PPAR-γ. Similar to our qRT-PCR and WB results, ICC analysis demonstrated that treatment with 25 μM of CK reduced the expression of C/EBPα and PPAR-γ (Figure 3). These results indicate that CK inhibits adipogenesis by restricting the expression of adipogenic markers including C/EBPα, FABP4, and PPAR-γ.

### 3.3. CK Inhibits ROS Production by Regulating the Expression of Anti-Oxidant Enzymes in Mature 3T3-L1 Adipocytes

An increase in intracellular reactive oxygen species (ROS) in adipocytes has been reported to be associated with obesity [7]. Therefore, we investigated the effect of CK on the activity of ROS during adipogenesis. We found a significant increase in ROS production in the differentiated group compared to the undifferentiated group (ND). We also found that treatment with the ROS scavenger NAC (5 mM) as well as 25 μM CK significantly inhibited differentiation-induced ROS generation in mature adipocytes (Figure 4A,B). 

Lipid droplets (LDs) are present in almost all cells and are organelles that regulate the storage and homeostasis of intracellular triglycerides and other triglycerides [34]. Abnormalities in the number or size of LDs influence the onset or progression of many metabolic diseases, including obesity, insulin resistance, and type 2 diabetes [33,35,36]. As shown in Figure 4C,D, the size and number of LDs were significantly increased in D than in ND, but the increased size and number of LDs was abolished by CK and NAC treatment.

Next, the effect of CK (0, 10, 20, 40 µM) treatment on the expression of ROS-related antioxidant enzymes during adipocyte differentiation was performed by qRT-PCR (Figure 4E) and WB (Figure 4F,G). We found that the mRNA expression measured in cells with 20 μM and 40 μM CK was significantly increased 10-fold and 30-fold for catalase and 5-fold and 10-fold for GR compared to D. The mRNA expression of GPx and SOD2 was significantly increased by 4-fold and 5-fold, respectively, by CK 40 µM treatment (Figure 4E).

We also observed that the protein expression of ROS scavenging enzymes, including catalase, GPx, GR, SOD2, and SOD3 increased in the CK-treated groups (20 μM or 40 μM) and the NAC-treated group compared to the control group (D) (Figure 4F,G). Taken together, these results indicate that the decrease in ROS production by increased expression of antioxidant enzymes including catalase, GPx, GR, SOD2 and SOD3 is involved in the reduction in CK-induced adipogenesis.

### 3.4. CK Inhibits the Mitotic Clonal Expansion (MCE) Process in 3T3-L1 Pre-Adipocytes

As the above analyses indicated that CK influences the early stage of adipocyte differentiation, we performed a flow cytometry analysis to determine if CK affects cell cycle progression during MCE. During the stage of differentiation, when pre-adipocytes reach confluence, cells enter the GO/G1 phase during which cell proliferation does not occur. When treated with DMI, a differentiation-inducing agent, the cell cycle was initiated, and cells began to proliferate. During differentiation, the number of cells was increased by two to three times, and the process of mitotic clonal expansion process started to begin.

When 3T3-L1 pre-adipocytes were treated with 0, 10, 20, 30, and 40 μM of CK for 48 h, cell accumulation increased by 24, 31, 39, 43, and 48% during the G2M phase, respectively (Figure 5A,B). To elucidate how CK affects cell cycle arrest, we investigated the protein expression of cell cycle regulators during the G2M phase. 3T3-L1 adipocytes treated with CK exhibited elevated levels of potent cell cycle inhibitor proteins p21 and p27 as well as a decreased expression of CDK1 and Cyclin B1 compared with control groups (Figure 5C–G). In the group treated with 40 µM of CK, the expression of cell cycle regulators had an effect compared to DMI, but it was not concentration-dependent. These results suggest that CK prevents the progression of the cell cycle by arresting cells in the G2M phase during the MCE stage of adipogenesis.

### 3.5. CK Inhibits AKT/ERK/P38 Signaling in Differentiation of 3T3-L1 Cells 

As the addition of differentiation medium inducer (DMI) during MCE increases the number of pre-adipocytes by two to three times, we counted 3T3-L1 pre-adipocytes at 0, 24, 36, and 48 h after treatment with DMI to determine whether CK affects MCE. After the DMI was applied, the number of 3T3-L1 precursor adipocytes was increased by 1.5 times after 24 h and 2.5 times after 48 h (Figure 6A). There was no statistically significant difference in the number of 3T3-L1 pre-adipocytes when 10 μM of CK was treated in combination with DMI over the 48 h period. However, treatment with 20, 30, and 40 μM CK prevented an increase in total cells in a statistically significant manner compared to the group only treated with DMI (Figure 6A). Our growth curve results indicate that CK inhibited mitotic clonal expansion during adipogenesis (Figure 6A).

Inducing adipogenesis via treatment with an DMI triggers the insulin- and IGF-1-activated phosphatidylinositol 3-kinase/protein kinase B (PI3K/PKB or AKT) as well as the mitogen-activated protein kinase/extracellular signal-regulated kinase (MAPK/ERK) pathways, which are involved in the formation of adipocytes during the early stages of differentiation [37,38]. Since CK blocked mitotic clonal expansion and the early phase of adipogenic differentiation, we performed a WB analysis to determine if CK affects the aforementioned signaling pathways. We observed that the phosphorylation of AKT, ERK, and p38 was upregulated in the differentiation-induced group compared to the undifferentiated group. However, we found that this elevated phosphorylation could be reduced through CK treatment (Figure 6B–E). These results suggest that CK may be involved in the regulation of adipogenesis by inhibiting the insulin-stimulated P38, ERK, and AKT-mediated signaling pathways. 

### 3.6. CK Inhibits Lipogenesis of 3T3-L1 Adipocytes via Suppression of PPAR-γ Expression

AMPK and its target, acetyl CoA carboxylase (ACC), are important energy sensors involved in the regulation of lipid metabolism and are key regulators of pre-adipocyte differentiation and adipogenesis. The activation of AMPK is known to be associated with decreased lipid storage [16]. The phosphorylation of AMPK and ACC was observed to be increased in the 25 μM CK-treated group compared to the differentiated control group. Specifically, the protein levels of p-AMPK and p-ACC were increased by 2.9-fold and 2.5-fold, respectively (Figure 7A–C). However, the expression of PPAR-γ, a key factor in adipogenesis, was significantly reduced by CK treatment (Figure 7A,D). We used the pharmacological AMPK inhibitor, Dorsormorphin (5 µM), to investigate if CK-mediated AMPK activation is directly involved in the inhibition of adipocyte differentiation. 

The increased phosphorylation of AMPK and ACC in mature adipocytes was inhibited by simultaneous treatment with CK and Dorsormorphin in a statistically significant manner. In addition, we found that the PPAR-γ protein level was significantly increased in the group treated with CK and Dorsormorphin than in the group treated with only CK (Figure 7A,E). In sum, our data strongly suggest that CK inhibits the overall adipogenesis process in 3T3-L1 adipocytes by regulating AMPK activation. 

## 4. Discussion

Rates of obesity are increasing worldwide. While several drugs such as lorcaserin, orlistat, phentermine, and topiramate have been used to treat obesity, they have been reported to cause serious side effects [39]. For this reason, complementary and alternative therapies using natural products with few side effects have been receiving increasing attention. In this paper, we investigated the effects of ginsenoside CK on the formation of adipocytes and explored the associated underlying mechanisms. Our results showed that CK significantly suppressed the differentiation and adipogenesis of 3T3-L1 cells, and it reduced the accumulation of lipid droplets and triglyceride content as well as inhibited intracellular ROS production. Furthermore, we established that the anti-adipogenic effect of CK is achieved by regulating the expression of C/EBPα, PPAR-γ, and FABP4 through the activation of AMPK and inhibition of ERK/p38 and AKT signaling pathways. Although the potential efficacy of ginseng and ginsenosides for obesity has been reported by many studies [26,40], researchers continue to search for substances that may be more effective. Ginsenoside CK, presented in this study, is a rare ginsenoside produced as a small amount by fermentation using β-glucosidase, and it has higher pharmacological activity than major ginsenosides such as Rb1, Rb2, Rd, Re, and Rg1 [40]. In addition, we determined that ginsenoside CK was safe to use, as our CCK-8 assay did not reveal any cytotoxic concentrations in 3T3-L1 pre-adipocytes after various concentrations of CK (10, 20, 40, 60, 80 μM) (Figure 1B). 

Obesity is a complex, multifactorial, and chronic disease caused by an imbalance between energy intake and expenditure. A persistent impairment of energy homeostasis leads to weight gain, which can lead to obesity, which is characterized by increased fat accumulation in adipose tissue [26]. An increase in adipose tissue mass results from an increase in the number (hyperplasia) and size (hypertrophy) of fat cells. The formation of lipid droplets in mature adipocytes occurs due to the accumulation of lipids, including triglycerides (TG) [1,2]. Therefore, methods to inhibit processes which increase the number of cells as well as the production of adipocytes have been suggested as important research objectives in order to alleviate dietary obesity and metabolic diseases. Our findings suggest that the reduction in lipid accumulation using CK treatment at an early stage of adipogenesis may be beneficial. In other words, non-cytotoxic concentrations of CK, applied during the initial stage of adipocyte differentiation, may be able to suppress adipogenesis by inhibiting lipid accumulation and triglycerides formation.

As aforementioned, adipocytes differentiation is controlled by various cellular pathways, and in particular, it depends on the sequential regulation of transcription factors responsible for the expression of genes related to adipocyte production. Transcription factors C/EBPα and PPAR-γ are key regulators of adipogenesis [17], and adipogenesis-specific genes such as SREBP-1c and FABP4 are regulated by C/EBPα and PPAR-γ during adipocyte differentiation [41,42]. Through our experiments, we found that the expression levels of mRNA and protein for the aforementioned transcription factors were decreased following treatment with CK in a concentration-dependent manner. In another report, it showed that ginsenoside CK (5 µM) inhibited the transcriptional expression of PPAR-γ, aP2 and C/EBPα and inhibited the adipogenesis in 3T3-L1 cells [30]. However, in our study, treatment with 10 µM of CK did not affect the transcriptional expression of ad-ipogenic genes. This discrepancy with the previous results is thought to be caused by the difference in the CK treatment period between the studied experimental conditions and the adipogenesis stage. 

One of the interesting things in our study is that the mRNA expression level of adiponectin was significantly decreased (Figure 2 and Figure 3). Adiponectin (also known as AipoQ) is an adipocyte-derived factor that has been shown to be associated with insulin resistance, type 2 diabetes, and obesity [43,44,45]. As low plasma adiponectin levels correlate with low adiponectin gene expression in adipose tissue [46,47,48,49], previous studies have suggested that low levels of adiponectin may be relevant in metabolic disorders such as obesity and insulin resistance. However, conflicting results have also been reported, which state that the overexpression of adiponectin in vitro enhances 3T3-L1 fibroblast proliferation and promotes adipocyte differentiation as well as increases lipid accumulation and insulin-responsive glucose transport in mature adipocytes [48]. In addition, adiponectin has been shown to increase lipid oxidation in skeletal muscle and muscle cells [49,50] as well as decrease hepatic glucose production in hepatocytes [51]. Studies have also reported that the administration of adiponectin to intact rodents improves their glucose tolerance and reduces plasma triglycerides [49,50]. In another study, rosiglitazone-treated mice exhibited significantly increased expression of adiponectin as well as enhanced adipocyte differentiation [52]. Therefore, the observed reduction in adiponectin following CK treatment may have occurred due to the anti-adipogenic effect.

Several studies have found that oxidative stress is associated with metabolic disorders, including obesity and diabetes [7,10,53]. Oxidative stress can be generated when the balance in the regulatory system between antioxidants and reactive oxygen species (ROS) is disrupted (ROS). Oxidative stress causes the uncontrolled production of free radicals, which contributes to obesity and various other complications [9]. It has been demonstrated that reactive oxygen species are by-products of cellular metabolism driven by multiple intracellular mechanisms and can cause damage to cellular biomolecules such as DNA, proteins and lipids when cells are continuously exposed to increased ROS [54,55]. The generation of ROS is also closely related to adipocyte differentiation and lipid accumulation [56]. In addition, increased levels of ROS during adipogenesis were associated with increased expression of PPAR-γ and an accelerated expansion of mitotic clonal expansion [57,58]. NAC is an antioxidant that reduces ROS, and the role of NAC in obesity has been reported in many studies [59]. A recent in vitro study showed that NAC attenuated biomarkers related to adipocyte differentiation pathways such as PPAR-γ and C/EBPβ in 3T3-L1 pre-adipocytes [60]. These effects have been shown to be associated with elevated glutathione levels through NAC supplementation [57,60]. It was also reported that the antioxidant NAC inhibits adipocyte differentiation through the activation of ROS-scavenging-related factors [58]. These anti-adipogenic effects are elicited by the increased levels of SOD, catalase, glutathione peroxidase and the activity of peroxynitrate, which serve to reduce ROS [61,62]. Various studies showed that ROS production is tightly regulated by ROS-generating enzymes, such as SOD, GPx, GR, and NADPH oxidase 4 (NOX4) [9,63]. Despite the presence of an active antioxidant system, the expression and activity of antioxidant enzymes such as SOD, GPX or catalase were decreased in WAT in obese mouse models, while ROS production was increased [10,63]. SOD converts superoxide (O_2_^−^) to hydrogen peroxide (H_2_O_2_) and O_2_, which is further broken down into water by catalase, GPx and peroxyredoxin (Prx) [56]. The various forms of mammalian SOD (SOD-1, -2, -3) are location-dependent, but their ability to neutralize ROS is common [64]. SOD has been reported to play an essential role in handling oxidative stress and ameliorating the obesity phenotype. A recent study showed that in vivo treatment with NanoSOD in mice fed a high-fat diet improved plasma triglyceride, hepatic triglyceride, and hepatic lipid accumulation levels [65]. It has been reported that SOD3 plasmid hydrodynamic injection treatment has smaller epididymal, inguinal, and perirenal white adipose tissues, and body weight is also improved in mice fed a high-fat diet compared to mice without SOD3 [66]. We observed that CK inhibited ROS generation and lipid accumulation during 3T3-L1 adipocyte differentiation, whereas it increased the expression of antioxidant enzymes such as catalase, GPx, GR, SOD2 and SOD3.

The adipogenic process can be induced using DMI, and growth-arrested pre-adipocytes re-enter the cell cycle for several cell divisions before adipogenic genes are expressed [19]. In that process, CDKs play an essential role in regulating cell re-entry into the cell cycle, and p21 and p27 serve as key cell cycle regulators in the G1 and S phases [28]. Our data show that CK treatment induces cell arrest in the G2M phase, decreases CDK1 and Cyclin B expression, and increases the expression of p21 and p27. Furthermore, CK may be involved in the inhibition of adipogenic differentiation by not only by inducing G2M phase arrest but also by inhibiting cell clonal expansion.

In recent years, many signaling molecules and pathways have been identified as important regulators of adipose tissue in obesity-related diseases. Among them, the most well-characterized are the mitogen-activated protein kinase (MAPK), AKT, and AMPK families [38,67]. In addition, the extracellular-signal-regulated kinase (ERK), c-Jun N-terminal kinase (JNK), p38, and MAPK signaling pathways are known to induce various intracellular responses via phosphorylation when activated by various extracellular stimuli [68]. In particular, ERK activation, which is involved in the cell cycle progression, is known to be essential to the induction of MCE and adipogenesis [69,70]. Several studies have reported that ERK differentiates pre-adipocytes into mature adipocytes by causing the increased expression of C/EBPα, C/EBPβ and PPAR-γ [69]. p38 MAPK promotes cell differentiation in the early stage of adipogenesis [71,72], and the AKT signaling pathway is also involved in adipogenesis [67]. AKT mediates the signaling pathways of insulin or IGF-I during adipogenesis, and the activation of AKT is required for insulin-induced glucose transport and energy metabolism in mature adipocytes [29]. Thus, activation of the AKT pathway in 3T3-L1 pre-adipocytes also appears to be involved in adipocyte differentiation [73]. An examination of how CK treatment affects the activation of signaling molecules that are essential for adipocyte differentiation will provide further clarity regarding the functional role of CK. In this context, it is quite significant that the phosphorylation of ERK, p38, and AKT were so meaningfully reduced in a concentration-dependent manner when mature adipocytes were treated with CK. Our findings suggest that ginsenoside CK suppresses adipogenic activity by downregulating the expression of adipogenesis-related transcription factors via the inhibition of MAPK and AKT phosphorylation.

AMPK is an important protein that maintains energy homeostasis by synthesizing and breaking down fatty acids and regulating energy metabolism pathways. When AMPK is activated, it induces sub-target ACC inactivation to attenuate lipogenesis by inhibiting expressions of FABP4, FAS, and SREBP-1c proteins [22,24,74]. Therefore, the activation of AMPK is an important factor when treating obesity and type 2 diabetes [24]. 

In this study, we found that CK treatment increased the phosphorylation of AMPK and target ACCs. In addition, CK treatment decreased the expression of PPAR-γ, which is a major adipogenic transcription factor, and its effect was reversed by treatment with dorsomorphine, which is a pharmacological inhibitor of AMPK. These results suggest that AMPK is directly involved in the CK-mediated inhibition on adipocyte differentiation, and that CK may regulate adipogenesis by activating AMPK. 

## 5. Conclusions

In summary, our study demonstrated that CK can inhibit the differentiation of 3T3-L1 pre-adipocytes without cytotoxic effects. Furthermore, CK suppressed not only the accumulation of lipid and triglyceride but also the production of ROS via inhibition of cell clonal expansion during the early stages of adipogenesis. We found that CK treatment regulated the expression of adipogenesis-related factors by activating the AMPK signaling pathway, and it inhibited adipocyte differentiation by suppressing the phosphorylation of ERK, p38, and AKT. In conclusion, the results of this study suggest that CK may have potential as a treatment or preventative tool for obesity and other obesity-related metabolic diseases. However, it is necessary to determine if CK is absorbed into the intestine, and has a direct effect on adipose tissues, by studying its bioavailability. 

## Figures and Tables

**Figure 1 antioxidants-11-01890-f001:**
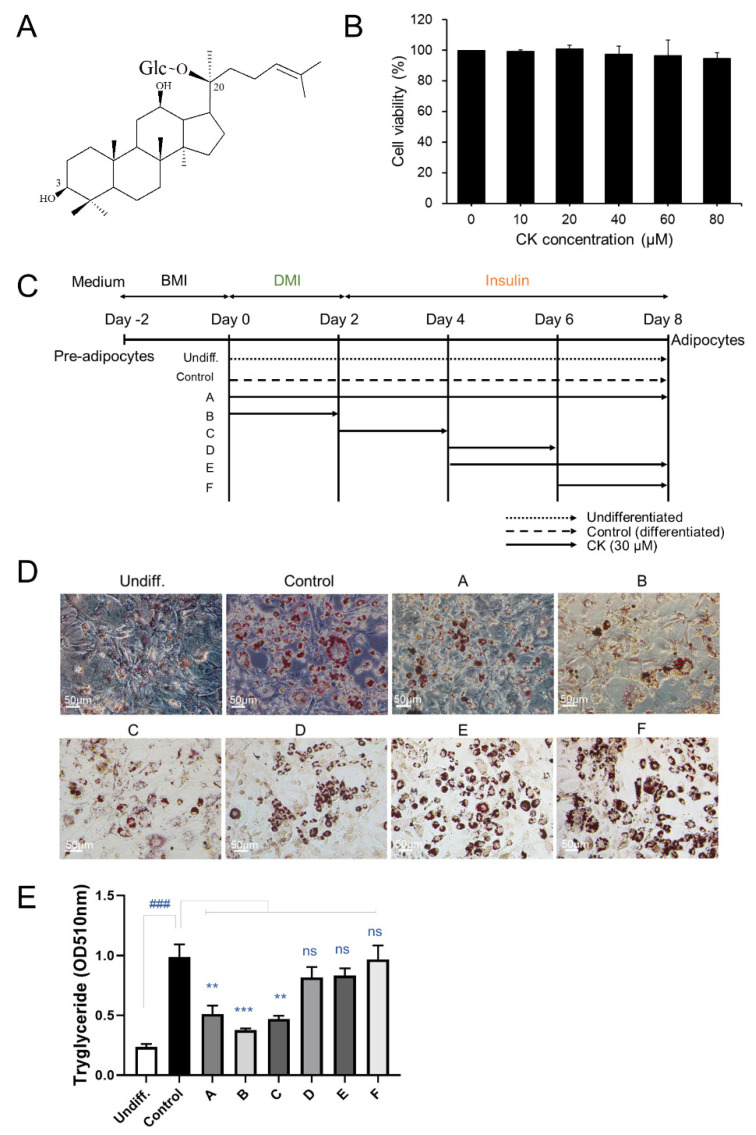
Effect of CK on cell proliferation and lipid accumulation in 3T3-L1 cells. (**A**) Chemical structure of compound K (CK). (**B**) 3T3-L1 pre-adipocytes were treated with CK at various concentrations (1, 10, 20, 40, 60, and 80 µM) for 48 h. (**C**) Schematic diagram of various CK treatment periods during adipocyte differentiation. (**D**) 3T3-L1 cells were stimulated with differentiation medium (DMI) and treated with 30 µM of CK at the indicated adipogenesis stages. After 8 days of differentiation, cells were stained with Oil Red O staining. (**E**) Stained Oil Red O was eluted with isopropanol, and absorbance was measured at 510 nm. Three independent experiments were performed, and the results were expressed as mean ± SEM. *^###^*: *p* < 0.01, significant difference between DMI group and undifferentiated group; **: *p* < 0.01 and ***: *p* < 0.001, significant difference between DMI group and DMI + CK treatment group. The results were considered not significant (ns) when *p* > 0.05.

**Figure 2 antioxidants-11-01890-f002:**
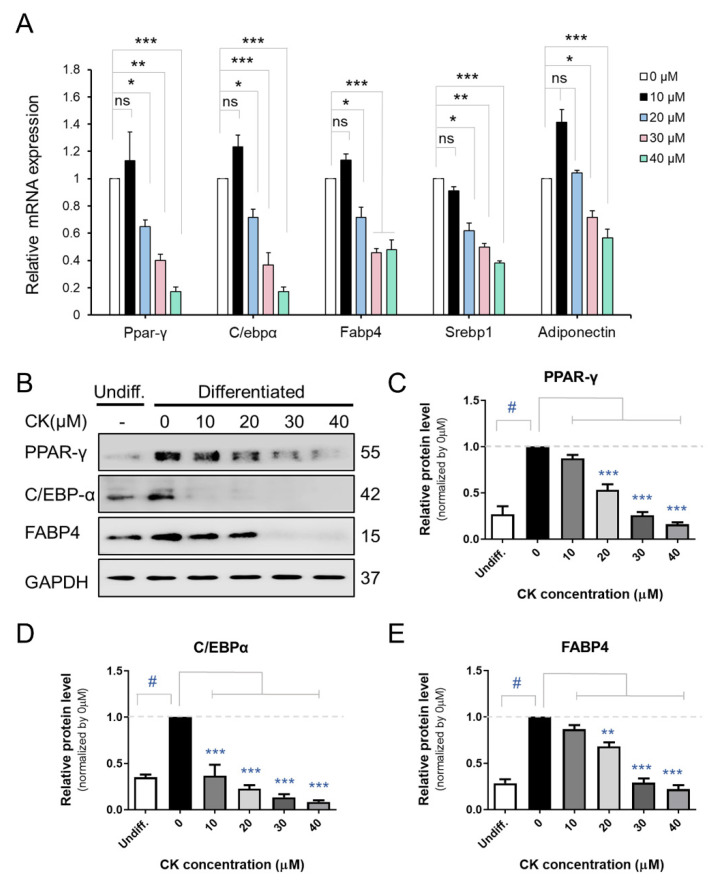
Effect of CK on adipogenesis in 3T3-L1 cells. (**A**) Cells were treated with various CK concentrations (0, 10, 20, 30, 40 µM) and underwent 8 days of differentiation. Figure (**A**) displays the mRNA expression of Ppar-γ, C/ebpα, Fabp4, Srebp1c, and Adiponectin genes in 3T3-L1 adipocytes. (**B**) Immunoblot analysis of PPAR-γ, C/EBPα, and FABP4 proteins according to CK concentration treatment during adipogenesis in 3T3-L1 cells. (**C**–**E**) Quantified and normalized protein levels of PPAR-γ (**C**), C/EBPα (**D**), and FABP4 (**E**), respectively. Data were represented with mean ± SD for *n* = 3. *: *p* < 0.05, **: *p* < 0.01, and ***: *p* < 0.001, significant difference between DMI group and DMI + CK treatment group; *^#^*: *p* < 0.001, significant difference between DMI group and undifferentiated group (Undiff.); DMI + CK 0 µM. The results were considered not significant (ns) when *p* > 0.05.

**Figure 3 antioxidants-11-01890-f003:**
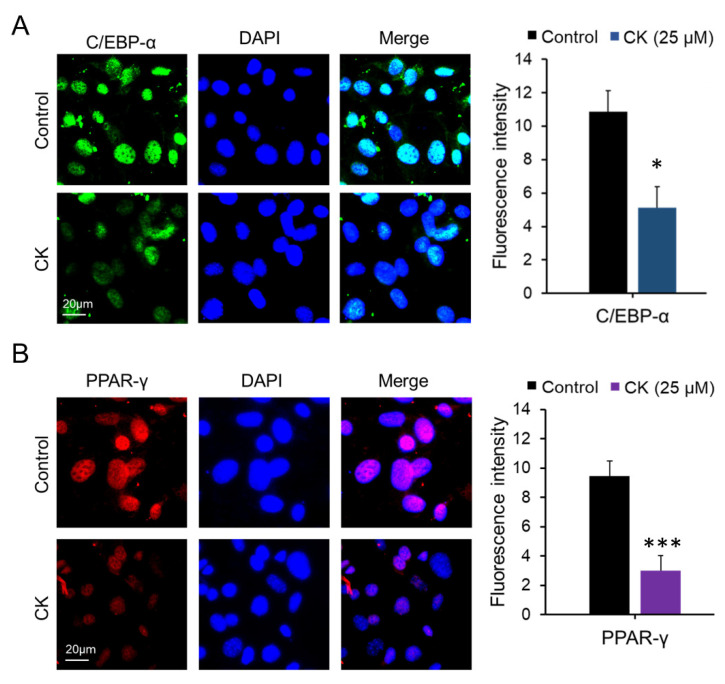
Effect of CK on the expression of key adipogenic proteins in 3T3-L1 cells. During 8 days of differentiation, 3T3-L1 cells were stimulated with 25 μM CK, and (**A**) C/EBPα and (**B**) PPAR-γ proteins were measured via immune cell staining (ICC). Fluorescence intensity was measured using Image J histogram software. The data were expressed as mean ± SD (*n* = 3) and analyzed using Student’s *t*-test. Control group: differentiated group; CK: CK treated group. *: *p* < 0.05, and ***: *p* < 0.001 compared to control. Control: differentiated group.

**Figure 4 antioxidants-11-01890-f004:**
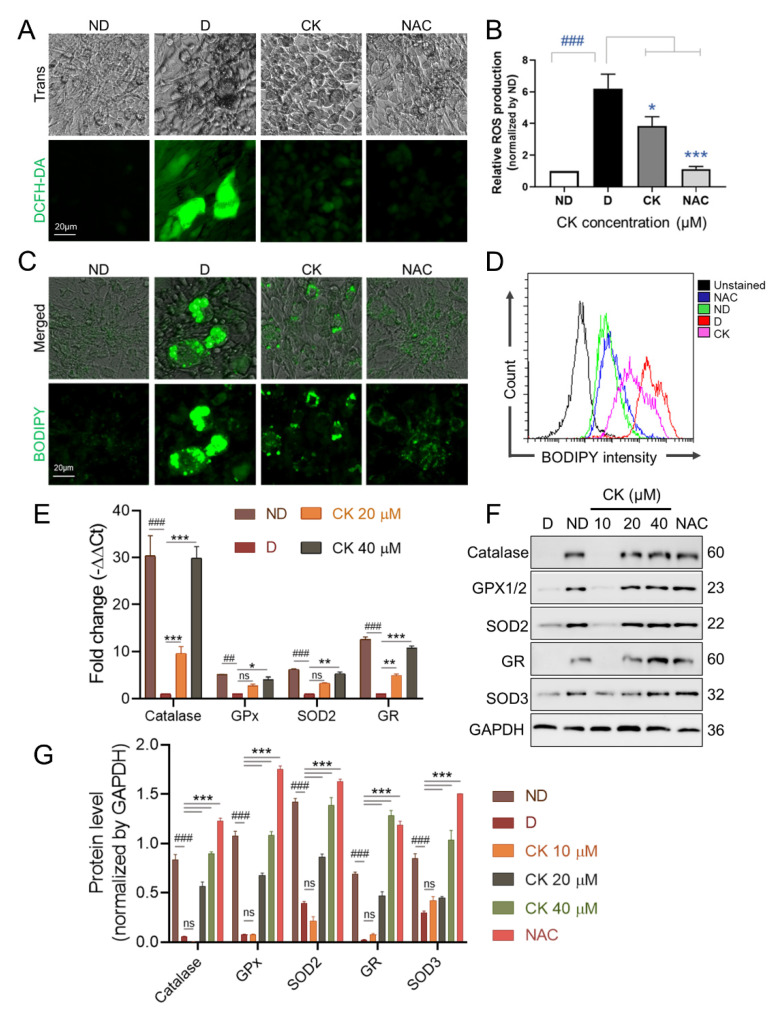
Effect of CK on ROS levels and expression of ROS-associated proteins in 3T3-L1 cells. The differentiation of pre-adipocytes was cultured for 8 days in differentiation medium with or without NAC 5 mM or various concentrations of CK (0, 10, 20, 40 µM). (**A**) ROS production was assessed using DCFH-DA dye in 3T3-L1 cells cultured for 8 days in differentiation medium with or without NAC 5 mM/25 µM CK. (**B**) DCF fluorescence intensity was measured at a wavelength of 490 nm, and relative DCF fluorescence intensity was expressed as a multiple of DCF fluorescence relative to ND. Data are presented as mean ± SD in the three experiments. *^###^*: *p* < 0.001 vs. ND and *: *p* < 0.05 vs. D, ***: *p* < 0.001 vs. D. (**C**,**D**) Fully differentiated 3T3-L1 cells in differentiation medium with or without NAC 5 mM/25 µM CK were stained with 2 µM BODIPY probe at 37 °C for 15 min. Stained LDs by BODIPY were observed under a fluorescence microscopy (**C**) and quantified by flow cytometry analysis (**D**). (**E**) The mRNA expression levels of the anti-oxidant enzyme genes Catalase, GPx, SOD2, and GR were evaluated by qRT-PCR using the 2^−ΔΔCT^ method. Data were obtained from at least 3 independent experiments; relative mRNA expression levels are expressed as fold change compared to D and values are expressed as mean ± SD. (**F**) Protein levels of antioxidant enzymes in fully differentiated 3T3-L1 cells in differentiation medium with or without 5 mM NAC or CK (0, 10, 20, 40 µM) were tested by Western blotting. (**G**) Protein band intensities were quantitatively analyzed using the Image J software. Glyceraldehyde 3-phosphate dehydrogenase (GAPDH) was used as RNA and protein loading control. Data error bars correspond to the mean ± SD of three independent experiments. *^##^*: *p* < 0.01, *^###^*: *p* < 0.001 vs. ND and *: *p* < 0.05, **: *p* < 0.01, ***: *p* < 0.001 vs. D. ND, undiff.; D, differentiated group. The results were considered not significant (ns) when *p* > 0.05.

**Figure 5 antioxidants-11-01890-f005:**
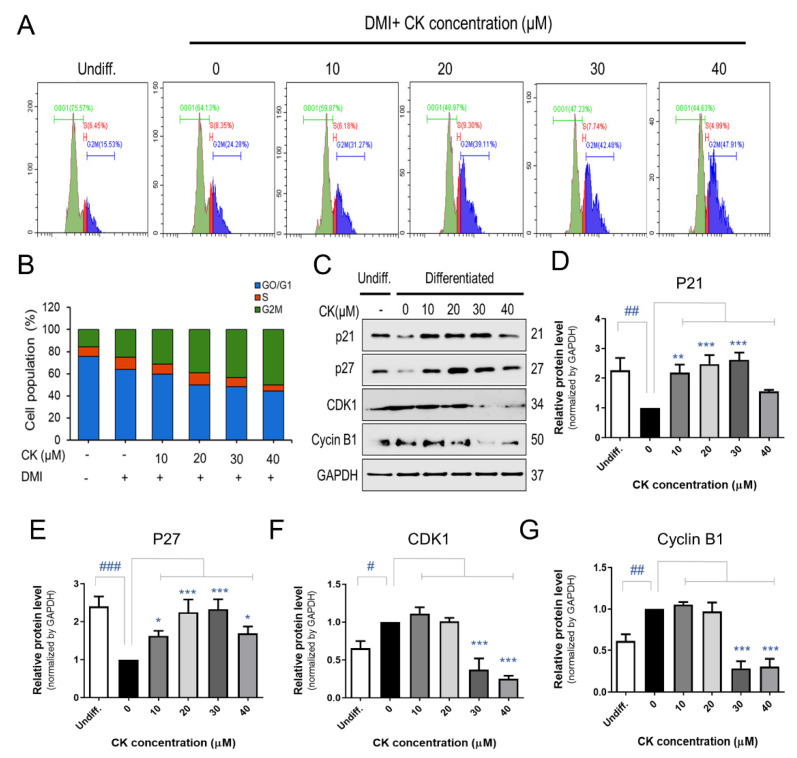
Effect of CK on mitotic clonal expansion (MCE) during adipogenesis in 3T3-L1 cells. (**A**) The effect of CK treatment on MCE during adipogenesis in DMI-induced pre-adipocytes was measured by performing a cell cycle analysis using flow cytometry. 3T3-L1 pre-adipocytes were treated with DMI and various concentrations of CK (10, 20, 30, 40 μM) for 48 h. (**B**) Quantification of (A): data are presented as the average value of three repeated experiments. (**C**) The protein levels of p21, p27, CDK1, and Cyclin B1 in cells stimulated with CK (10, 20, 30, 40 μM) for 48 h. Normalized protein levels (regarding the samples analyzed and depicted in Figure 5C) for (**D**) p21, (**E**) p27, (**F**) CDK1, and (**G**) Cyclin B1. The results are mean ± SD from three independent experiments. ^#^: *p* < 0.05, ^##^: *p* < 0.01, ^###^: *p* < 0.001 compared to Undiff. group; *: *p* < 0.05, **: *p* < 0.01, ***: *p* < 0.001 compared to Differentiated group without CK treatment (0 µM CK).

**Figure 6 antioxidants-11-01890-f006:**
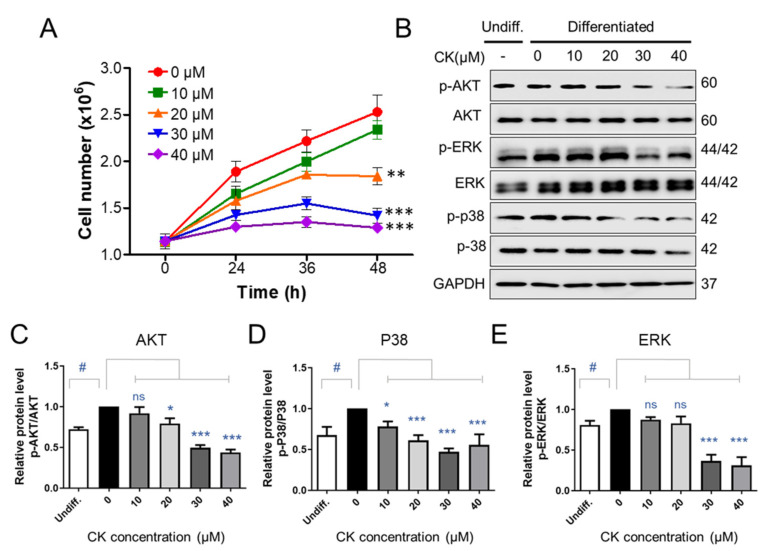
The effect of CK on signaling molecules involved in mitotic regulation. (**A**) Two days after confluence, 3T3-L1 cells were incubated with DMI medium, with and without various concentrations of CK, for indicated times. The number of cells at the indicated points (0, 24, 36, and 48 h) after DMI-induced differentiation was counted using a Luna cell counting slide. The experiment was repeated independently three times. **: *p* < 0.01, ***: *p* < 0.001 compared to 0 µM CK group. (**B**) Protein levels of p-AKT, AKT, p-ERK, ERK, p-P38, and p38 during 48 h of differentiation were analyzed by immunoblot assay using specific antibodies. GAPDH was used as a loading control. (**C**–**E**) Quantitative results of protein levels of Akt (**C**), p38 (**D**), and ERK 1/2 (**E**) adjusted to each total protein levels. Protein band intensities were quantified by densitometry using Image J software. ^#^: *p* < 0.05 vs. Undiff. group, *: *p* < 0.05, ***: *p* < 0.001 vs. to Differentiated group without CK treatment (0 µM CK). The results were considered not significant (ns) when *p* > 0.05.

**Figure 7 antioxidants-11-01890-f007:**
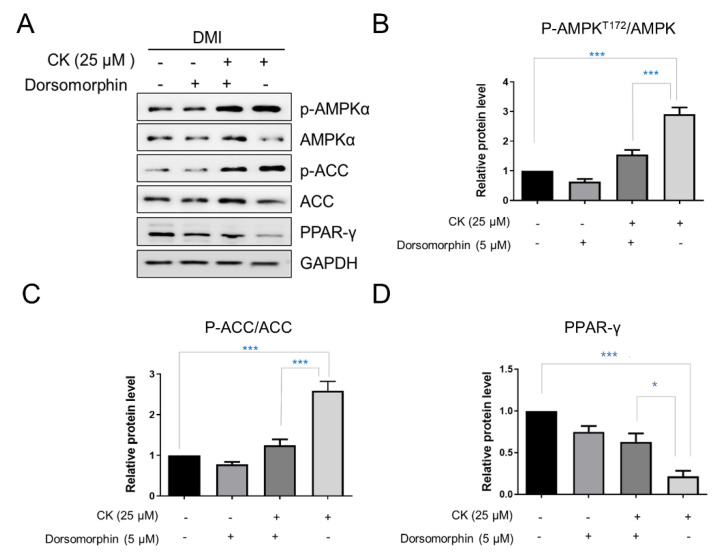
Effect of CK on AMPK signaling pathway. Confluent 3T3-L1 cells, with and without 5 µM Dorsormorphin, were treated with CK and differentiation was induced for 8 days. (**A**) Expression of p-AMPK, AMPK, p-ACC, ACC, PPAR-γ was analyzed by Western blotting. (**B**–**D**) Relative protein levels of AMPK (**B**), ACC (**C**), and PPAR-γ (**D**) were measured via densitometry and analyzed using Image J software. Data were expressed as mean ± SD from triplicated experiments. *: *p* < 0.05, and ***: *p* < 0.001, significantly different from the differentiated group.

**Table 1 antioxidants-11-01890-t001:** Primer sequences used for quantitative RT-PCR analysis.

Gene	Accession Number	Forward Primer (5′-3′)	Reverse Primer (5′-3′)
*Gapdh*	NM_001411845.1	CTCGTGGAGTCTACTGGTGT	GTCATCATACTTGGCAGGTT
*Ppar-* *γ*	U01664.1	CAGCCTTTAACGAAATGACCA	TGTGGAGTAGAAATGCTGGA
*C/ebp* *α*	NM_001287523.1	AAACAACGCAACGTGGAGA	GCGGTCATTGTCACTGGTC
*Fabp4*	NM_001409513.1	TGATGATCATGTTAGGTTTGGC	TGGAAACTTGTCTCCAGTGAA
*Srebp1*	AF374266.1	CTGGTCTACCATAAGCTGCAC	GACTGGTCTTCACTCTCAATG
*Adipoq*	NM_009605.5	GGAGAGAAAGGAGATGCAGGT	CTTTCCTGCCAGGGGTTC
*Catalase*	M62897.1	TCC GGG ATC TTT TTA ACG CCA TTG	TCG AGC ACG GTA GGG ACA GTT CAC
*Gpx*	NM_001329527.1	GGGCAAGGTGCTGCTCATTG	AGAGCGGGTGAGCCTTCTCA
*GR*	BC057325.1	CACGACCATGATTCCAGATG	CAGCATAGACGCCTTTGACA
*SOD2*	NM_013671.3	GGGTTGGCTTGGTTTCAATA	AGGTAGTAAGCGTGCTCCCA

*Gapdh*, glyceraldehyde 3-phosphate dehydrogenase; *Ppar-**γ*,Peroxisome proliferator-activated receptor gamma; *C/ebp**α*, CCAAT enhancer binding protein alpha; *Fabp4*, Fatty acid-binding protein 4; *Srebp1c*, Sterol regulatory element binding protein 1c, *GPx*; Glutathione peroxidase, *GR*; Glutathione reductase, *SOD2*; Superoxide dismutase 2.

## Data Availability

The data are contained within this article.

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
