# Peer review of "Ginsenoside CK Inhibits the Early Stage of Adipogenesis via the AMPK, MAPK, and AKT Signaling Pathways"

_antioxidants, 2022, doi:10.3390/antiox11101890_

Round 1

Reviewer 1 Report

In this paper, the molecular strategy that can treat the severity of obesity and obesity, and the relationship between obesity and oxidative stress are well summarized in the introduction to suggest the justification for the study. It is logical to study CK's effect on stages of adipocyte maturation.

After confirming the cytotoxicity of CK, this paper discusses the overall effect of adipogenesis in 3T3-L1 preadipocytes and suggests AMPK, MAPK, and AKT signaling as a mechanism for revealing that CK is involved in early stage of adipogenesis. This is very interesting.

However, the following 6 things need to be supplemented.

1.       In Figure 2, CK was treated for 8 days. At the low dose (10 mM), the expression of adipogenesis-related genes at the protein level is increased compared to the control group. This shows the same trend at the mRNA level. Supplementary explanations and arrangements are needed for this. Also, the experimental group is different at the mRNA level and the protein level, but I am curious as to why the 'undiff' group was not added at the mRNA level.

2.       It would be good to unify the form of putting the scale bar in the figures of Figure 3 and Figure 4.

3.       Figure 5 shows that CK inhibits the progression of the cell cycle of 3T3-L1 preadipocytes and prevents mitotic clonal expansion. When adipogenesis-related gene expression was confirmed in Figure 2, the expression was decreased in a dose-dependent manner, but the expression of P21, P27, CDK1, and Cyclin B1 was not dose-dependent. Therefore, when reading this part, the reader may wonder about the other roles of these factors through CK besides the mechanism of CK inhibiting early stage of adipogenesis through inhibition of mitotic clonal expansion through AMPK, MAPK, and AKT signaling. Relevant explanations need to be supplemented. Since Figure 8 mentions the AMPK pathway, it would be helpful to provide a direct correlation between the Figure 5 and Figure 7 results.

4.       The typo in Figure 7D (PPPAR-γ) should be corrected.

5.       In Figure 7, the expression of ACC and PPARγ was confirmed through AMPK, which plays a larger role in adipogenesis than the factor that inhibits the initial stage of adipogenesis. Therefore, in the discussion, the sentence 'we also demonstrated that the anti-adipogenic effect of CK is achieved by regulating mitotic clonal expansion during adipogenesis via the AMPK/AKT/ERK/P38 pathway' is ambiguous. There is a need to separately describe the AMPK pathway or to additionally present the initial steps of AMPK-related adipogenesis-related gene expression.

6.       The topic of the thesis and Figure 4A may be interpreted as having no correlation. Therefore, the relationship between obesity and antioxidant effects should be discussed in more detail in the discussion. Supplementary explanations are needed for the expected clinical effects of CK-induced oxidative stress amelioration by adipogenesis.

Reviewer 2 Report

In the present study Oh and Chung aimend to investigate the effects of compound K (CK) on the stages of adipocyte maturation, as well as the signaling pathways involved in this process. It was concluded that CK exerts anti-adipogenic effects by regulating the early stages of mitotic clonal expansion during adipocyte differentiation via the AMPK, AKT, and ERK/p38 signaling pathways. I have the following comments:

- Abstract: Please define ROS, GPx, CDK, G2M, AKT and ACC.

- Introduction should be shortened.

- Authors should state a hypothesis.

- It is not clear how the Western immunoblotting experiments were carried out. It seems from the Figures (2B, 4F, 5C, 6B and 7A) that one membrane was used to obtain all the results for each figure since only one representative of GAPDH was shown. If this is the case, it should be clearly stated that the membranes were debloted and then incubated with another antibody. This procedure is not appropriated when there are proteins with high levels of aminoacid homology such as those from Fig. 6B and 7A (phospo and total) and 4F (SOD 2 and SOD3).   

- Authors should comment the limitations of using DCFH-DA for ROS detection. Although it is not selective, this dye is usually used for H2O2 detection (combined with other methods). This is relevant to the study since CK inhibited ROS generation, while increased the expression of antioxidant enzymes such as catalase, GPx, SOD2 and SOD3. SOD may affect H2O2 generation since it converts O2- into H2O2 ans catalase and GPx may affect H2O2 levels since they catalises the transformation of H2O2 into H2O and O2.   

- It is not apropriated to used SD for such a small sample size (n=3). Please re-analyse using SEM.

- What is the origin of ROS? (mitochondrial, NADPH oxidase...)

- How CK exerted its effects on ROS? (direct scavenging, reduction of protein expresion, increased of antioxidant defense....).

Author Response

Thank you very much for reviewer’s helpful comments. We tried to edit the manuscript according to reviewer’s comments. We believe reviewer’s comment made our manuscript improved. Please find our response (blue-typed) to reviewer’s specific comments (black-typed) below.

Reviewer #2

In the present study Oh and Chun aimed to investigate the effects of compound K (CK) on the stages of adipocyte maturation, as well as the signaling pathways involved in this process. It was concluded that CK exerts anti-adipogenic effects by regulating the early stages of mitotic clonal expansion during adipocyte differentiation via the AMPK, AKT, and ERK/p38 signaling pathways. I have the following comments:

Main points:

Point 1.  Abstract: Please define ROS, GPx, CDK, G2M, AKT and ACC.

Response 1. Thanks for your comment. We defined each abbreviation in the abstract, as the reviewer pointed out. We hope the reviewer understand that the reason we were unable to define all abbreviations in the abstract was due to the word limit.

Point 2. Introduction should be shortened.

Response 2. Thanks for your comments. As the reviewer pointed out, we tried to keep the introduction part short. We marked the deleted parts by tracking in the introduction. However, please understand that if the introduction is too short, other reviewers may request supplementation again. We believe that the reviewer can figure out our efforts.

Point 3. Authors should state a hypothesis.

Response 3. Thank you for your comment. We fully understood the reviewer’s point to state the hypothesis of our study. However, our study is to reveal a new mechanism of ginsenoside CK on adipogenesis rather than to discover any new type of ginsenoside or its function. In that respect, to state a hypothesis is not appropriate to our manuscript. Several previous studies have reported that CK has an inhibitory effect on adipogenesis in 3T3-L1 cells. However, the effect of CK on the stage of adipocyte differentiation or the specific mechanism of CK has not yet been clearly elucidated. Therefore, we sought to elucidate the exact molecular mechanism of CK on adipogenesis in this manuscript. For that reason, we mentioned the purpose of our experiment rather than a hypothesis at the end of the introduction (page 3, line 110). We hope that the reviewer understand this situation.

Point 4. It is not clear how the Western immunoblotting experiments were carried out. It seems from the Figures (2B, 4F, 5C, 6B and 7A) that one membrane was used to obtain all the results for each figure since only one representative of GAPDH was shown. If this is the case, it should be clearly stated that the membranes were debloted and then incubated with another antibody. This procedure is not appropriated when there are proteins with high levels of aminoacid homology such as those from Fig. 6B and 7A (phospo and total) and 4F (SOD 2 and SOD3).  

Response 4. Thank you for your comment. We understand what the reviewer pointed out, but it is misunderstanding of our data. In our experiments, we performed Western blotting more than 3 times for each target. Then, each density was obtained from the entire experimental blots, and a representative image was selected and inserted into the figures of manuscript. Therefore, as the reviewer pointed out, not all figures have the same GAPDH. However, only GAPDH shown in Figure 5 and Figure 6 is the same, since both experiments were conducted under the same conditions. Nevertheless, we replaced the GAPDH in Figure 6 with a new band image, as the reviewer pointed out. We believe that the reviewer can easily understand this explanation by looking at the WB original image file that has already been submitted.

Point 5. Authors should comment the limitations of using DCFH-DA for ROS detection. Although it is not selective, this dye is usually used for H2O2 detection (combined with other methods). This is relevant to the study since CK inhibited ROS generation, while increased the expression of antioxidant enzymes such as catalase, GPx, SOD2 and SOD3. SOD may affect H2O2 generation since it converts O2- into H2O2 ans catalase and GPx may affect H2O2 levels since they catalises the transformation of H2O2 into H2O and O2.  

Response 5. Thank you for your good comment. We fully agree with the reviewer’s opinion. As the reviewer points out, it is true that DCFH-DA is limited in its ability to detect all types of reactive oxygen speices (ROS). For that reason, various methods are being used to detect or label individual ROS types. However, in our paper, DCFH-DA method was used to determine the content of intracellular reactive oxygen species (ROS), not individual oxygen pieces such as H2O2, NO2 • , CO3 •− or (O2 •−). Furthermore, H2O2 acts as an inducer to generate ROS, and DCFH-DA is not commonly used to detect H2O2 by itself.

These can be found in the references below (especially in Ref. 1). Please understand that DFCH-DA is still widely used for ROS detection in many papers, even though it has some limitations like weak fluorescence signals and high background. The purpose of our study is not to verify all individual levels of reactive oxygen species, and it was not possible to use all kinds of methods due to technical issues. Based on the reviewer’s point, we will actively refer to it in our next studies.

  • Ref 1. Murphy M.P. et al., Guidelines for measuring reactive oxygen species and oxidative damage in cells and in vivo. Nat Metab. 2022. 2022 Jun;4(6):651-662. PMID: 35760871
  • Ref. 2) Zhu C. et al., No evident dose-response relationship between cellular ROS level and its cytotoxicity – a paradoxical issue in ROS-based cancer therapy. Sci Rep. 2014 May 22;4:5029. PMID: 24848642

Point 6. It is not appropriated to used SD for such a small sample size (n=3). Please re-analyse using SEM.

Response 6. Thank you for your comment. We have thoroughly discussed what the reviewer points out, and we fully understood it. However, we would like to say that we are skeptical about using standard error mean rather than standard deviation. In our experiments, we performed at least three WB analyses. In addition, we would like to say that the most suitable analytic method for such WB results is standard deviation. Please refer to the provided website for reference (https://www.licor.com/documents/7bd2dev0rfjofad7dr1zf85wip4g4it3). This site shows the protocol for “Quantitative Western Blot Analysis with Replicate sample”. In this file (page 10), it is written as follows:

  • Bar graphs are more suitable for large data sets. Many journals currently discourage the use of error bars that indicate the standard error of measurement (SEM). Error bars that indicate the standard deviation of your measurements may be more appropriate (Ref 1, Ref 2, Ref 3).
  • Ref 1. Krzywinski M, Altman N. Error bars. Nat Meth. 10(10): 921-22 (2013).
  • Ref 2. Fay DS, Gerow K. A biologist’s guide to statistical thinking and analysis. WormBook, ed. The C. elegans Research Community, WormBook, doi/10.189/wormbook.1.159.1, http://www.wormbook.org (2013).
  • Ref3. Nagele P. Misuse of standard error of the mean (SEM) when reporting variability of a sample. A critical evaluation of four anaesthesia journals. Br J Anaesth. 90: 514-16.

Point 7. What is the origin of ROS? (mitochondrial, NADPH oxidase...)

Response 7. Thank you for your nice question. The origin of ROS affected by CK in 3T3-L1 cells may be relevant to mitochondrial ROS. The reason we reached this conclusion is that we have already performed experiments on mitochondrial ROS and mitochondrial activity with CK treatment.  Unfortunately, the current manuscript does not contain the following: This is because we are currently preparing for submission to other journals with different types of results. However, we would like to answer the question by presenting the undisclosed results as follows. Although we cannot add to this manuscript, here we provide reviewer with the results of Mitosox for mitochondrial ROS and Mitotracker stain for mitochondrial activity. We hope that the reviewer can fully understand our efforts and situation.

Point 8. How CK exerted its effects on ROS? (direct scavenging, reduction of protein expression, increased of antioxidant defense....).

Response 8. Thank you for your question. In our results, the effect of CK on ROS was shown to inhibit the formation of intracellular ROS when CK was treated with 25 µM in Figure 4, which is similar to the result when treated with NAC, a ROS scavenger. Therefore, it was confirmed that there is a positive correlation in the formation of ROS and the generation of lipid droplets.

Several studies have reported that increased ROS production in obesity is associated with NADPH oxidase and anti-oxidant enzymes. In our results, we did not see NOX4, the ROS source enzyme, but we confirmed that CK increased the protein expression of the ROS-scavenging enzymes SOD2, GPx, GR, and catalase. These results can be explained that CK inhibits adipocyte differentiation by increasing the expression of endogeneous anti-oxidant enzymes.

Round 2

Reviewer 2 Report

Significant changes were made in the manuscript.

I believe that in the present form the manuscript should be accepted for publication.